# In Situ Observation of Microstructural and Inclusions Evolution in High-Strength Steel Deposited Metals with Various Rare Earth Pr Contents

**DOI:** 10.3390/ma15031257

**Published:** 2022-02-08

**Authors:** Tianli Zhang, Weiguang Wang, Yiming Ma, Naiwen Fang, Sanbao Lin, Zhuoxin Li, Sindo Kou

**Affiliations:** 1School of Materials Engineering, Shanghai University of Engineering Science, Shanghai 201620, China; wwg14579@163.com; 2Harbin Welding Institute Limited Company, Harbin 150028, China; myiming246@sina.com (Y.M.); naiwen20@163.com (N.F.); 3State Key Laboratory of Advanced Welding and Joining, Harbin Institute of Technology, Harbin 150001, China; sblin@hit.edu.cn; 4College of Materials Science and Engineering, Beijing University of Technology, Beijing 100124, China; zhxlee@bjut.edu.cn; 5Department of Materials Science and Engineering, University of Wisconsin, Madison, WI 53706, USA; kou@engr.wisc.edu

**Keywords:** high-strength steel, rare earth Pr, deposited metal, in situ observation, microstructure, mechanical properties

## Abstract

The evolution of austenite, acicular ferrite, upper bainite and martensite, and the nucleation of inclusions in the microstructure of high-strength steel deposited metals, was systematically investigated using three kinds of A5.28 E120C-K4 metal-cored wires with various rare earth Pr contents. Grain structure evolution in the process of high temperature, dispersoid characteristics of inclusions and the crystallographic characteristics of the microstructure were assessed. Compared with no addition of Pr_6_O_11_, adding 1%Pr_6_O_11_ resulted in refined, spheroidized and dispersed inclusions in the deposited metal, leading to an increase in the pinning forces on the grain boundary movement, promoting the formation of an ultra-fine grain structure with an average diameter of 41 μm. The inclusions in the deposited metals were Mn-Si-Pr-Al-Ti-O after Pr addition; the average size of the inclusions in the Pr-containing deposited metals was the smallest, while the number and density of inclusions was the highest. The size of effective inclusions (nucleus of acicular ferrite formation) was mainly in the range of 0.6–1.5 μm. In addition, the content of upper bainite decreased, while the percentage of acicular ferrite increased by 24% due to the increase in the number of effective inclusions in the Pr-containing deposited metals in this study. This study shows that the addition of 1% Pr_6_O_11_ is efficient in achieving fine interlaced multiphase with an ultrafine-grained structure, resulting in an enhancement of the impact toughness of the deposited metal.

## 1. Introduction

The use of 690–1200 MPa high-strength steel is widespread in industries with extremely high safety requirements, such as the petrochemical industry, nuclear power, aerospace and rail transit. However, the corresponding welding consumables with matched strength and toughness are few, which limits its further application. In the welding of high-strength steel, the higher the strength level of the deposited metals, the more likely the occurrence of hydrogen embrittlement. At present, the microstructure of high-strength steel deposited metal can be optimized by adjusting the composition and proportion of alloying elements, so that its strength and toughness can be improved. Therefore, it is extremely important to study the effects of alloying elements on the microstructure and mechanical properties of high-strength steel deposited metals and the development of matched welding consumables [1,2,3].

Currently, the research on improving the strength and toughness of high-strength steel deposited metal mainly focuses on microalloying elements such as Nb, Ti, Cu, Ni and Zr. Fine inclusions can promote the formation of acicular ferrite (AF); inhibit the growth of bainite, ferrite side-plate, and proeutectoid ferrite in the austenite grains; affect the diffusion of other elements and the nucleation and growth of new phases and induce the change in microstructure and properties of the high-strength steel deposited metals [4,5,6,7,8,9]. During the welding process, the original austenite grains of the deposited metal with high carbon content grow rapidly, and a large amount of upper bainite (UB) and a small number of AF can be formed during the cooling process, so as to reduce the mechanical properties of deposited metals. In addition, the properties can be improved by preheating before welding, and a reduction in heat input and post-welding heat treatment [10,11,12,13]

Rare earth elements can spheroidize and refine inclusions in high-strength steel deposited metals, which acts as the pinning of grain boundaries to restrain grain coarseness. In the cooling process, rare earth inclusions can induce the formation of AF, thereby improving the toughness of deposited metals. Therefore, the mechanical properties of high-strength deposited metal can be improved by controlling the content of rare earth elements to refine grains, and by controlling the quantity, size, shape and composition of inclusions to promote the formation of AF [14,15,16].

Most current studies focus on high-strength steel deposited metal microstructures after solidification, but have not dealt with the real-time observation of the dynamic transformation of austenite, ferrite, bainite and martensite in high-strength steel deposited metals at high temperature. In this research, the starting point of phase transformation was not defined, and the grain nucleation growth as well as migration were not observed. The use of a high-temperature laser scanning confocal microscope for in situ observation of microstructure at high temperature can solve the above problems [6,17]. In this paper, in situ observation was carried out using a high-temperature confocal microscope to study the effect of Pr_6_O_11_ on the growth behaviors of austenite, ferrite, bainite and martensite in the microstructure of 800 MPa high-strength steel deposited metals. The chemical composition, microstructure, mechanical properties and nucleation of inclusions in the deposited metals were also discussed. The aim of this paper is to study the effect of Pr on the deposited metals so as to enrich and improve the theory on the effect of rare earth in welding metallurgy.

## 2. Materials and Methods

Three kinds of self-developed high-strength steel metal-cored wires (1.2 mm in diameter), designated as wire nos. 1–3, were used to prepare deposited metals by welding according to the AWS standard A5.28 E120C-K4 [18]. The addition of Pr_6_O_11_ was 0% weight percent with wire no. 1, 1% weight percent Pr_6_O_11_ with wire no. 2, and 2% weight percent Pr_6_O_11_ with wire no. 3. The welding parameters are shown in Table 1. The chemical composition of the deposited metals are shown in Table 2. The chemical compositions of deposited metals were determined with a Q4 optical emission spectrometer. The mechanical testing was conducted to ASTM A370-2019 standards. After preparing the standard specimens, the mechanical properties of the deposited metals were tested with a WAW-6000 tensile test machine, and the yield strength, tensile strength and elongation were recorded. The Charpy V-notch impact test was performed by an JB30B impact testing machine after the impact specimens were cooled to −40 °C.

From the position parallel to the surfacing surface on each deposited metal, 6 cylindrical specimens were cut to 6 mm in diameter and 3 mm in length. One of the specimens was ground and polished to be the metallographic specimen etched by 5% nital. The microstructures of the deposited metals were observed by scanning electron microscopy (SEM) with a Hitachi S-3400N microscope, and the inclusions in deposited metals were analyzed by energy dispersive spectrometer (EDS) analysis. Selecting the impact sample for metallographic analysis, a sample was cut with a thickness of 5 mm and a length and width of 10 mm × 10 mm below the fracture surface. After water grinding and polishing, the surface of the sample had no oxide layer and continuous corrosion pits. The surface of the sample was electropolished with chloric acid alcohol solution to eliminate the surface processing strain layer. The operating voltage during electrolysis was 12 V and the time was 15 s. The orientation relationship and crystallographic grain size of microstructure was examined by 20 KV voltage electron backscatter diffraction (EBSD) with a Tescan LYRA3 XMH SEM. The remaining 5 specimens were polished to be the high temperature metallographic specimens and mounted in an alumina crucible. The in situ observation was carried out by high-temperature laser scanning confocal microscopy with a Lasertec VL2000DX-SVF17SP and an Yonekura infrared image furnace. The specimens were heated to 1500 °C at a rate of 5 °C/s and then cooled at a rate of 5 °C/s to form austenite, ferrite, bainite and martensite microstructures. The photographs were taken at a speed of 15 images per second during the simulated thermal cycle. The OLYCIA-M3 quantitative analysis software was used to perform quantitative statistical analysis on the microstructure and inclusions of deposited metals.

## 3. Results and Discussion

### 3.1. Grain Growth Behavior during the Transformation of Austenite

The in situ observation of austenite grain growth in the deposited metal of high-strength steel without Pr_6_O_11_ is shown in Figure 1. The inclusions agglomerate in the deposited metal as shown in Figure 1a. As the temperature decreases, small austenite grains appear, as shown in Figure 1b. Small grain (grain no. 1) aggregates and swallows the surrounding austenite grains (grains nos. 2–3), to form a large grain (grain no. 4). During the nucleating process of austenite with the addition of 1% Pr_6_O_11_ as shown in Figure 2, the inclusions become dispersed in the grain boundaries and act as a pinning to prevent the extension of austenite grains in different directions, due to the refining effect of the rare earth element Pr on the inclusions. As a result, the growth of austenite grains was hindered, and the grains are significantly refined. With an addition of 2% Pr_6_O_11_ as shown in Figure 3, the quantity of inclusions is significantly reduced. The austenite grains nucleate first and then extend in different directions. Due to the lack of inclusion pinning, the growth of austenite grains is not restricted, so the grains are obviously coarsened.

From the statistical results of the average austenite grain size in Figure 4, it can be seen that with the increase in Pr_6_O_11_, the average austenite grain size first decreases and then increases.

When Pr_6_O_11_ was not added, the average size of the austenite grains reached 129 μm, because the growth of the grains was not restricted due to the agglomeration of inclusions. When 1% Pr_6_O_11_ was added, the addition of rare earth elements refined and spheroidized the inclusions, making them dispersed in the deposited metal. The inclusions have a pinning effect on the austenite grain boundaries, so they limit the growth of austenite grains [19]. The average size of austenite grains was 41 μm. When 2% Pr_6_O_11_ was added, the decrease in oxygen content in the deposited metal led to the reduction of nucleation sites of inclusions, which made the growth of austenite unrestricted, and the average size of austenite grains increased to 53 μm.

The −40 °C impact toughness of the depositded metal is shown in Figure 5. With the addition of Pr_6_O_11_, the impact toughness increased at first and then decreased. When 1% Pr_6_O_11_ was added, the impact toughness reached the maximum of 72 J.

Figure 6 shows the tensile strength and yield strength curves of the deposited metals with the addition of various Pr_6_O_11_ contents. With the addition of Pr_6_O_11_ from 0% to 2%, the strength of weld metal gradually decreased. With the addition of 2% Pr_6_O_11_, the tensile strength of the weld metal decreased from the initial value of 843 MPa to 814 MPa, and the yield strength decreased from 793 MPa to 720 MPa. With the addition of Pr_6_O_11_, the elongation of the weld metal first increased and then decreased. The 0% Pr_6_O_11_ had the lowest elongation, which was 17%. The 2% Pr_6_O_11_ had the highest elongation, which was 20%. The −40 °C impact toughness of the deposited metal is shown in Figure 2. With the addition of Pr_6_O_11_, the impact toughness increased at first and then decreased. When 1%Pr_6_O_11_ was added, the impact toughness reached the maximum of 72 J.

### 3.2. In Situ Observation of the Formation of AF, UB and Martensite

The in situ observation of the growth process of ferrite, bainite and martensite with different Pr_6_O_11_ content is shown in Figure 7, Figure 8 and Figure 9. It can be seen from Figure 7 that when Pr_6_O_11_ is not added, AF nucleates and grows on the inclusions, and UB nucleates and grows at the austenite grain boundary and then gradually occupies the space for austenite. Usually, the nucleation temperature of acicular ferrite in weld metal is 40–70 °C higher than that of bainite [20]. Therefore, acicular ferrite preferentially surrounds intragranular inclusions in the cooling process, thereby inhibiting the bainite transformation that can subsequently occur at prior austenite grain boundaries. In addition, the formed acicular ferrite interface can also stimulate the formation of secondary acicular ferrite. When the acicular ferrite laths nucleate and grow on the inclusions, the increase in the austenite/ferrite interface area can promote the nucleation of acicular ferrite, and the subsequent acicular ferrite nucleation provides a new austenite/ferrite interface, and such excited nucleation can make the distribution of acicular ferrite formed within the austenite grains relatively uniform [1]. The quantity of inclusions in the austenite grains is reduced due to the agglomeration of inclusions at the grain boundaries, so the growth of UB is not restricted, which leads to an increase in the proportion of UB in the austenite and poor impact toughness of the deposited metal. Figure 8 shows that when 1% Pr_6_O_11_ is added, AF in the grain boundaries first nucleates and grows on the inclusions, divides the austenite grains into many smaller and separated regions, and limits the subsequent formation of UB and martensite growth. As a result, the percentage of AF in the grains is greater than that of UB and martensite, thereby optimizing the impact toughness of the deposited metal. Figure 9 shows that when 2% Pr_6_O_11_ is added, the inclusion nucleation particles are reduced due to the decrease in oxygen, resulting in a decrease in the AF nucleation rate, so UB and martensite grow unrestricted. As a result, the content of UB and martensite in the grain boundary is higher, and the impact toughness of the deposited metal is reduced.

Figure 10, Figure 11 and Figure 12 show the microstructure and quantitative statistical analysis of deposited metals with different Pr_6_O_11_ content. It can be found that the content of AF first increases and then decreases with the addition of Pr_6_O_11_. When Pr_6_O_11_ is not added, the highest UB content is 46%, and the lower AF content is 28%. The average length of plates of UB and martensite is 51 μm and 46 μm. When 1% Pr_6_O_11_ is added, the highest percentage of AF is 52%; while the content of UB and martensite is the lowest, 21% and 27%, the average length of plates of UB and martensite is 32 μm and 28 μm. This is the because the addition of 1% Pr_6_O_11_ promotes the nucleation and growth of AF, inhibits the growth of UB and martensite, and generates the formation of an interlaced multiphase microstructure. AF preferentially nucleates and separates the austenite, so that the subsequently formed UB and martensite do not grow too large to realize grain refinement. When 2% Pr_6_O_11_ is added, the content of AF decreases to 29%, the content of UB and martensite increases and the average length of plates of UB and martensite is 53 μm and 39 μm. The microstructure of the deposited metal is not uniform, and the lath structure increases. Figure 13 is the schematic diagram of the growth process of the ferrite, bainite and martensite in deposited metals with different Pr_6_O_11_ contents. It can be seen from Figure 13a that when Pr_6_O_11_ is not added, inclusions in the grains are reduced due to the accumulation of inclusions in the grain boundary, so the nucleation sites of AF are reduced. Thus, the growth of UB and martensite is not restricted. As the content of UB and martensite increases, the toughness reaches as low as 45 J. Figure 13b shows that when 1% Pr_6_O_11_ is added, the inclusions are dispersed in the austenite grain boundary, and AF nucleation sites increase. A large amount of AF nucleates and grows in grain boundaries, and divides the austenite, so the subsequent growth of UB and martensite is restricted, and an interlaced multiphase microstructure is formed. The impact toughness of the deposited metal is improved to 72 J. It can be seen from the schematic diagram in Figure 13c that when an excessive amount of 2% Pr_6_O_11_ is added, the decrease in inclusions leads to a reduction of AF nucleation in austenite grains, resulting in the unrestricted growth of UB and martensite. As the UB and martensite increase, the impact toughness of the deposited metal is reduced to 61 J. With the addition of Pr_6_O_11_ from 1% to 2%, the proportion of AF decreases from 52% to 29%, and the proportion of UB and M increases from 27% and 21% to 38% and 33%, respectively, resulting in a decrease in toughness. However, the addition of Pr_6_O_11_ has little effect on the tensile strength of the deposited metal, which is about 810–840 MPa.

Since the solid phase transition during the welding process is continuous and complex, the microstructure of the deposited metal is complex and uneven. As the temperature of the deposited metal changes, AF nucleates and grows prior to UB and martensite at high temperature, and divides the original austenite grains into several parts. The subsequently generated UB and martensite can only grow in the segmented region. In the formed interlaced multiphase microstructure, each grain has various orientations and the size of the grain is small. The cracks need to change many times when passing through this microstructure. Passivation or branching occurs at the large-angle grain boundary formed between UB and AF. More energy is consumed, so the deposited metal has excellent impact toughness. The grain size of the original austenite has been determined at high temperature. The more the divided structure of the complex phase is generated by the subsequent phase transformation, the more complex the degree of interlacing, and the more excellent strength and toughness of the deposited metal. UB and martensite act as the main strengthening phase, and AF acts as the main toughening phase. It can be seen from Figure 11a that the microstructure of the deposited metal without adding Pr_6_O_11_ is basically a single orientation. However, Figure 11b shows that with the addition of 1% Pr_6_O_11_, the deposited metal has more interlaced multiphase microstructures with different orientations. This explains why the impact toughness of deposited metal is significantly higher with 1% Pr_6_O_11_ than without Pr_6_O_11_.

AF is considered to be an excellent microstructural component that improves toughness by effectively refining grains. AF grains can divide the coarse austenite grains into small single regions, forming a mixed structure of fine particles. The microstructure of AF has larger angled grain boundaries, which can effectively increase the crack propagation path during fracture. The increase in the energy required for crack propagation helps to improve the low temperature impact toughness of the weld metal, which may explain why the impact energy of deposited metal without Pr_6_O_11_ is higher than that of deposited metal with 1%Pr_6_O_11_. The addition of Pr_6_O_11_ can serve as the core of AF heterophasic nucleation. The grain core preferentially attaches to the surface of these impurities and promotes the formation of a large amount of AF, which inhibits the formation of phases such as martensite and UB. Martensite and UB are the main strengthening phases, which can increase the strength and hardness of the weld metal. AF is the toughening phase in the weld metal, which mainly improves the low temperature impact toughness of the weld metal [16]. The formation of AF reduces the martensite and UB strengthening phases in the weld metal, so the addition of Pr_6_O_11_ leads to a decrease in strength and an increase in toughness.

Therefore, Pr_6_O_11_ mainly improves the toughness of deposited metal through the following three aspects: firstly, rare earth Pr promotes the nucleation of AF and inhibits the growth of UB and M; secondly, rare earth inhibits the growth of original austenite grains during the welding; thirdly, the deposited metal forms an interlaced multiphase microstructure with different orientations. Rare earth elements can improve the mechanical properties of deposited metal, mainly by changing the aggregation status of inclusions, making inclusions dispersed in the deposited metal to pin the austenite grain boundary and limit the growth of austenite. It also helps to promote the nucleation and growth of AF and divide the austenite, so that the subsequent growth of UB and martensite is restricted, forming an interlaced multiphase microstructure and improving the impact toughness of the deposited metal. However, the amount of Pr_6_O_11_ added should be controlled. Excessive addition will reduce the oxygen content in the deposited metal, reduce the content of inclusions and AF nucleation sites, thereby reducing the impact toughness of the deposited metal of high-strength steel.

### 3.3. Inclusions in Deposited Metals

Figure 14, Figure 15, Figure 16 and Figure 17 show the SEM images, high-temperature laser scanning confocal microscopy images, EDS analysis and size statistics of inclusions in the deposited metals with different Pr_6_O_11_ content. Without adding Pr_6_O_11_, the inclusions are mainly composed of Fe, O, Si, Mn, Ti, Al and Zn. The inclusions are mainly Mn-Si-Zn oxide. The inclusion size is mainly distributed between 1.2–1.8 μm; the average size of the inclusion is 1.725 μm and 0.89% in area. The inclusions agglomerate irregularly, which is not conducive to the formation of AF [21,22].

When 1% Pr_6_O_11_ is added, the inclusion is mainly composed of Fe, O, Si, Mn, Ti, Al, Pr and Cr, and the size is mainly distributed in the range 0.6–1.5 μm; the average size of the inclusion is 1.206 μm and 0.92% in area. Zhang et al. [20] pointed out that most of the inclusions used as AF nucleation were between 0.4 and 1.6 μm in size, and they were composite inclusions containing multiple elements. Therefore, adding 1% Pr_6_O_11_ inclusion size is conducive to AF nucleation and growth. In addition, the inclusions present a dispersive distribution, are small in size and their shape is spherical or elliptical. This indicates that the addition of 1% Pr_6_O_11_ promotes the refinement and spheroidization of the inclusions. Pr was found in the deposited metal and the inclusions are cored with Si-Al-Ti-Pr oxides and covered by an outer layer of Mn-Fe oxides. It was found that the inclusions in the deposited metal changed from the long strip of a larger size to the spherical or elliptical of a smaller size [23].

When 2% Pr_6_O_11_ is added, the inclusion mainly consists of Fe, O, Si, Mn, Ti, Al, Pr and Cr. The inclusion size is mainly distributed in the range 1.5–2.1 μm; the average size of the inclusion is 1.813 μm and 0.63% in area. The inclusions are cored with Al oxides and covered by an outer layer of Mn oxides. The O content in the inclusions is reduced from 1.18% to 0.90%, and the quantity of inclusions is significantly reduced. This is because that the deoxidation of the rare earth element Pr leads to a decrease in O content in the inclusions, and the formation of inclusions in the deposited metal is closely related to the O content [24]. The addition of rare earth affects O content in the deposited metal, which in turn affects the formation of inclusions. Therefore, the excessive addition of Pr_6_O_11_ may inhibit the formation of inclusions in the deposited metal; the decrease in inclusions leads to a reduction in AF nucleation in austenite grains, resulting in the unrestricted growth of UB and martensite.

The main mechanism of AF nucleation is the formation of an element depletion zone around inclusions. In the Pr-free deposited metal, only a small amount of AF grains nucleate on some inclusions. The inclusion consists of Al, Ti, Mn and O elements. It is observed that the inclusions are complex oxides of Al and Ti wrapped in MnO, which usually lead to the formation of a poor Mn zone due to the precipitation of MnO. The fundamental mechanism of AF nucleation in Pr-free deposited metal is the depletion of Mn near inclusions. The formation of a manganese-poor zone increases the phase transition temperature and nucleation driving energy, thus promoting the nucleation of AF grains. Compared with the Pr-free deposited metal, there are more AF grains nucleated in the inclusions in 1%Pr_6_O_11_, and the inclusions are characterized by composite oxides of Al and Ti covering MnO and Pr_6_O_11_. The presence of Pr_6_O_11_ in the vicinity of complex inclusions has contributed to the formation of high density of AF grains [9]. When 2% Pr_6_O_11_ is added, the inclusion mainly consists of Fe, O, Si, Mn, Ti, Al, Pr and Cr. The inclusion size is mainly distributed between 1.5–2.1 μm.

The addition of Pr_6_O_11_ in the deposited metal will promote the transformation of inclusions and form rare earth composite inclusions. With the addition of Pr_6_O_11_ in the deposited metal, the inclusion phases in spherical forms normally have the maximum surface tension under the premise of the same volume. The spherical inclusions have the minimum thermodynamic driving force during their growing process in the weld pool [25]. Thus the inclusions in the deposited metal change their shape.

With Al_2_O_3_ in the inclusions, the addition of rare earth will promote the formation of uniform Pr-Al-O rare earth composite inclusions [26,27]. During the cooling process of deposited metal, Pr_6_O_11_ reacts with Ti_2_O_3_ and Al_2_O_3_ inclusion particles to form rare earth composite inclusions. The diameter of the inclusions gradually increases, and the core of the inclusions transforms into Mn-Si-Pr-Al-Ti-O composite inclusions. The existence of Si and Mn in the inclusions is caused by the absorption of Si and Mn in the base metal and welding wire during the growth of inclusions. Mn in the inclusions is also an important factor affecting AF nucleation [28,29,30,31].

### 3.4. Crystallographic Characteristics of Microstructure in Deposited Metal

Figure 18 is the EBSD orientation maps of the microstructures of deposited metals with different Pr_6_O_11_ content. Figure 18a,b is the inverse pole image of the subunit structure of the microstructure. It can be seen from the inverse pole image that the austenite grains are divided into several orientations, and the microstructures of different orientations are intertwined. This intertwined structure is beneficial to suppress crack propagation and improve the toughness of the material. Figure 19 is a statistical diagram of the size and angle of grain boundaries in the microstructure of deposited metals with different Pr_6_O_11_ contents. The size of grains without Pr_6_O_11_ is mainly distributed in the range 4–8 μm. With adding 1% Pr_6_O_11_, the grain size becomes smaller, mainly distributed in the range 3–5 μm; with a small angle grain boundary (0°–15°), the quantity of grains is significantly reduced, and the number of large-angle grain boundaries (grain boundaries greater than 15°) is increased. When the crack crosses the large-angle grain boundary and enters adjacent grains, it inevitably changes the direction of crack propagation. Crack propagation will be hindered as it consumes energy during the process, so the impact toughness of deposited metal will increase [32]. The energy of a small-angle grain boundary is low, and the dislocation structure of the grain boundary is simple, so cracks can pass through easily, therefore reducing the impact toughness of the deposited metal. AF is considered to be an excellent microstructural component that improves toughness by effectively refining grains. AF grains can divide the coarse austenite grains into small single regions, forming a mixed structure of fine particles. The microstructure of AF has larger-angled grain boundaries. Therefore, with the addition of Pr_6_O_11_, the proportion of acicular ferrite increases so that the high-angle grain boundaries in the deposited metal increase. The reasons for the improvement in impact toughness of the deposited metal with the addition of 1% Pr_6_O_11_ include: firstly, the grain size is reduced, and the grains become refined; secondly, the quantity of small-angle grain boundaries decreases significantly, and crack propagation is hindered as a result of the increase in large-angle grain boundaries.

## 4. Conclusions

Compared with no addition of Pr_6_O_11_, adding 1%Pr_6_O_11_ can change the inclusion aggregation from agglomeration at the grain boundary to dispersed distribution within austenite grains. The size of the inclusions becomes smaller, and their shapes are spherical or elliptical. The inclusions pin the original austenite grain boundaries so as to effectively inhibit the growth of austenite grains. The size of inclusions is mainly distributed in the range 0.6–1.5 μm, which is conducive to the formation of AF nucleation particles. AF nucleates and grows on the inclusions, divides the austenite grains and restricts the growth of UB and martensite, forming an interlaced multiphase microstructure.

After adding 1%Pr_6_O_11_ in the deposited metals, the average size of austenite grains was at a minimum (41 μm). Small-angle grain boundaries were reduced and large-angle grain boundaries increased. The tensile strength of the deposited metal reached 834MPa, and the maximum impact toughness reached 72 J.

The inclusions in the deposited metal were Mn-Si-Pr-Al-Ti and Mn-Si-Pr-Al-Ti-O in the free-Pr and Pr contents of the deposited metals. The average size of inclusions decreased after Pr addition. These composite inclusions effectively induced the formation of AF.

With the addition of 2% Pr_6_O_11_, excessive rare earth elements caused the austenite grains to coarsen; the inclusion size was mainly distributed in the range 1.5–2.1 μm. AF nucleation sites were reduced, thereby reducing AF. The growth of UB and martensite were unrestricted, resulting in a reduction in the impact toughness of the deposited metal to 61 J.

## Figures and Tables

**Figure 1 materials-15-01257-f001:**
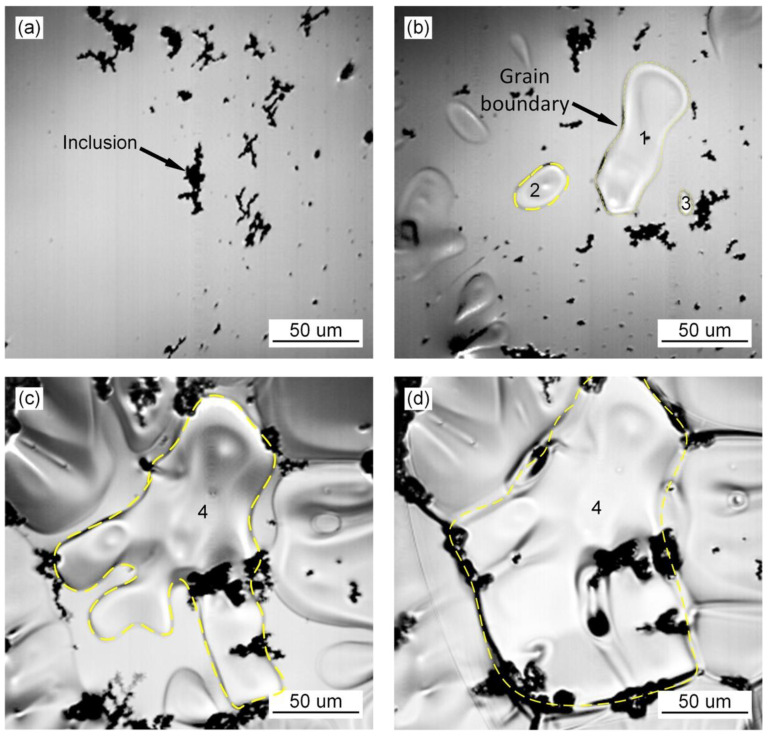
In situ observation of growth process of austenite at 0% Pr_6_O_11_: (**a**) 369.4 s, 1480.7 °C; (**b**) 373.59 s, 1460.5 °C; (**c**) 379.97 s, 1423.6 °C; (**d**) 386.95 s, 1378.3 °C.

**Figure 2 materials-15-01257-f002:**
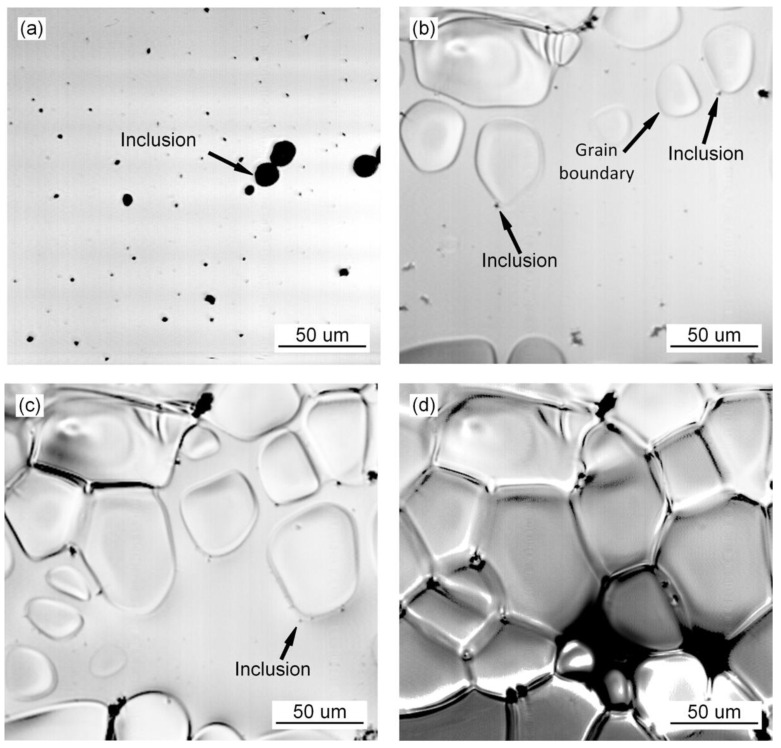
In situ observation of growth process of austenite at 1% Pr_6_O_11_: (**a**) 227.11 s, 1545.3 °C; (**b**) 308.51.92 s, 1424.7 °C; (**c**) 310.71 s, 1410.7 °C; (**d**) 314.10 s, 1387.2 °C.

**Figure 3 materials-15-01257-f003:**
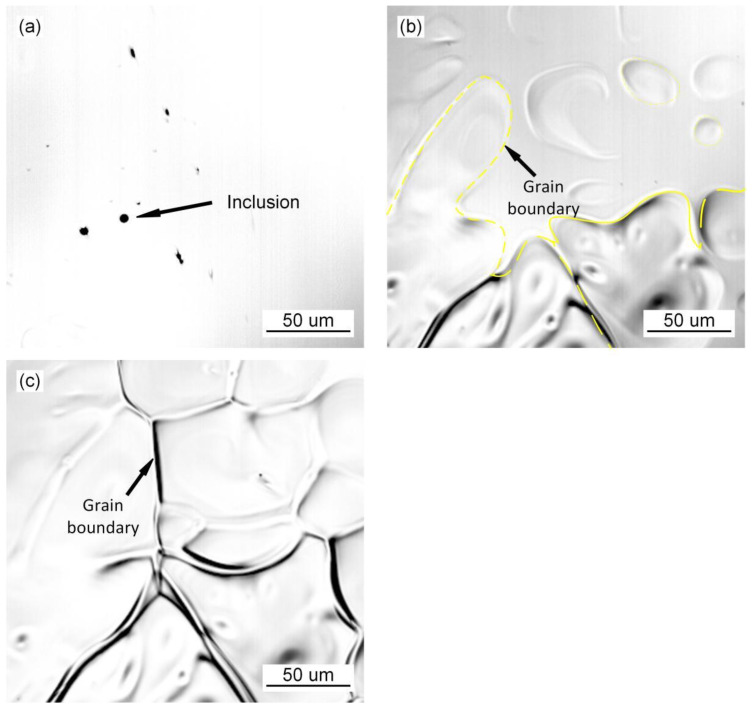
In situ observation of growth process of austenite at 2% Pr_6_O_11_: (**a**) 229.51 s, 1486.1 °C; (**b**) 235.9 s, 1459 °C; (**c**) 236.4 s, 1450.9 °C.

**Figure 4 materials-15-01257-f004:**
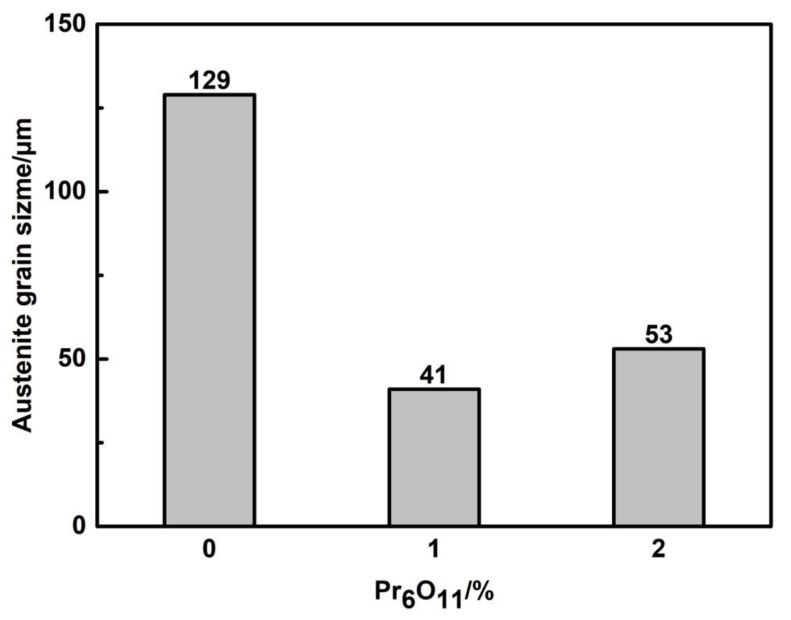
Average size of austenite grains with different Pr_6_O_11_ content.

**Figure 5 materials-15-01257-f005:**
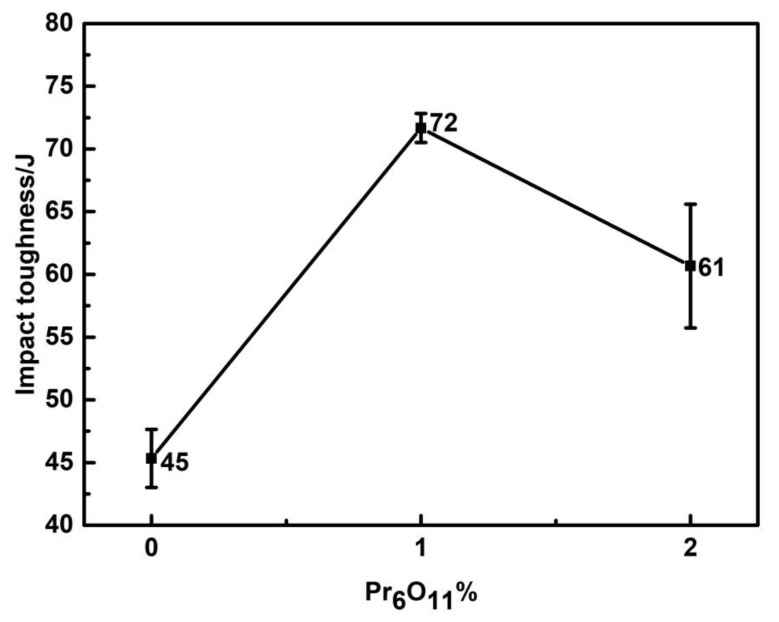
Effect of Pr_6_O_11_ on the impact absorbed energy of the deposited metals.

**Figure 6 materials-15-01257-f006:**
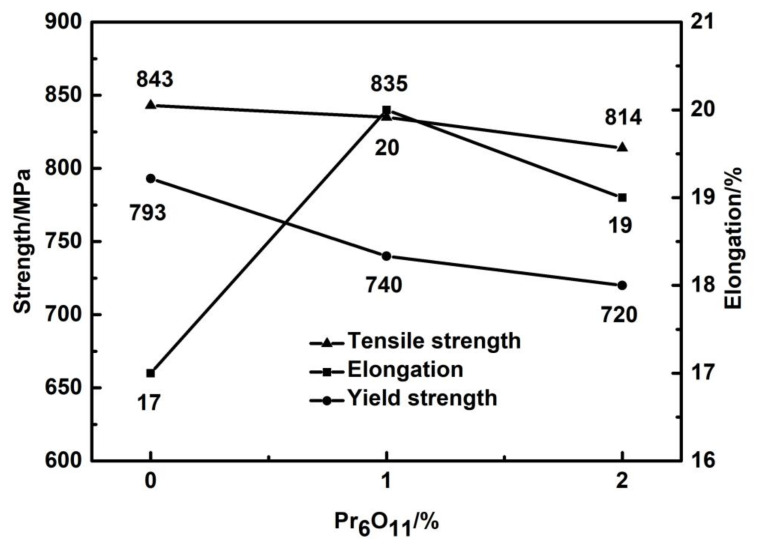
Effect of Pr_6_O_11_ on the strength and elongation of the deposited metals.

**Figure 7 materials-15-01257-f007:**
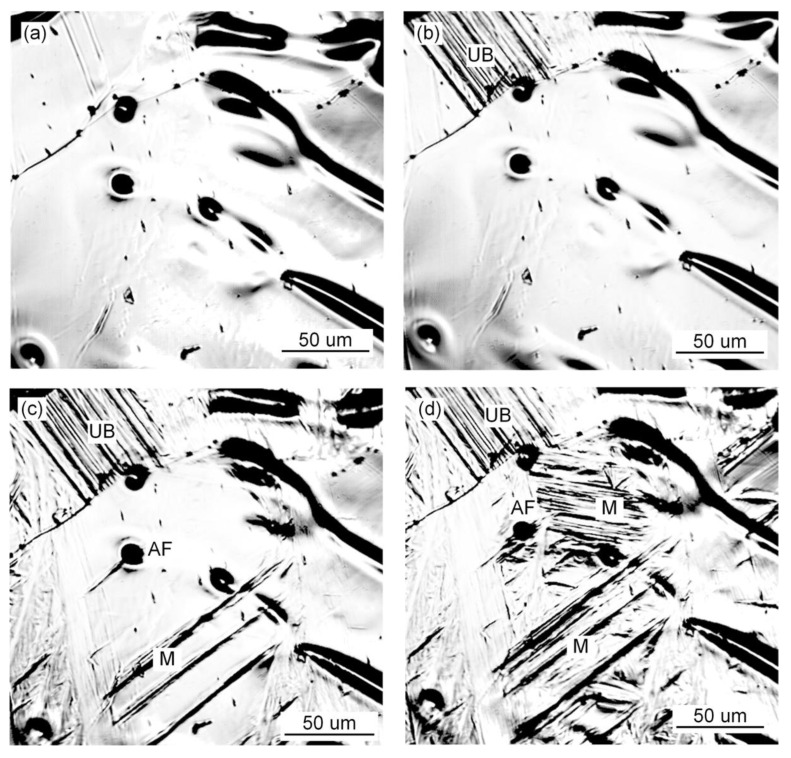
In situ observation of growth process of ferrite, bainite and martensite at 0% Pr_6_O_11_: (**a**) 887.93 s, 537 °C; (**b**) 890.92 s, 523 °C; (**c**) 893.51 s, 511 °C; (**d**) 897.30 s, 497.8 °C.

**Figure 8 materials-15-01257-f008:**
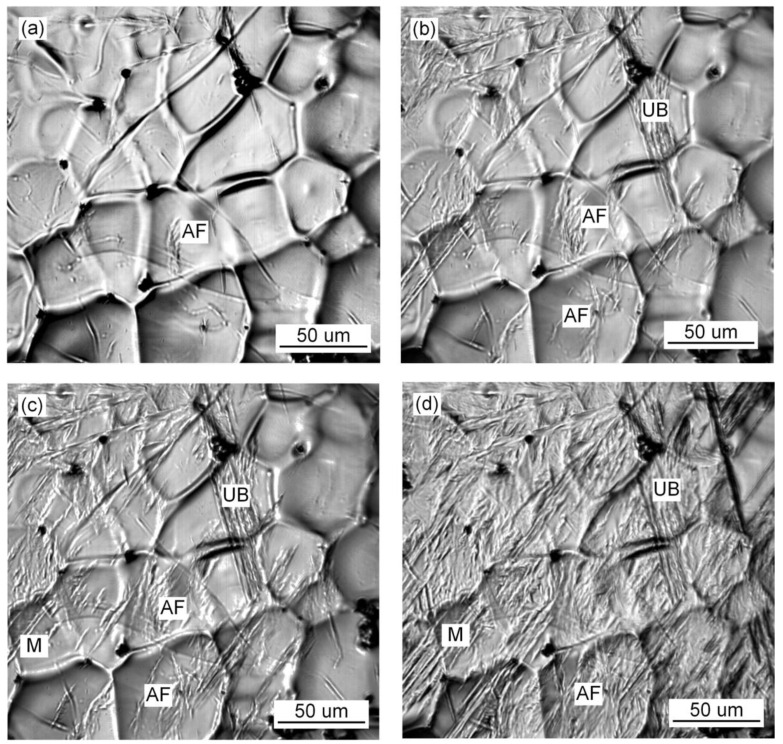
In situ observation of growth process of ferrite, bainite and martensite at 1% Pr_6_O_11_: (**a**) 480.9 s, 549.3 °C; (**b**) 482.68 s, 537.4 °C; (**c**) 483.68 s, 530.9 °C; (**d**) 486.47 s, 516.1 °C.

**Figure 9 materials-15-01257-f009:**
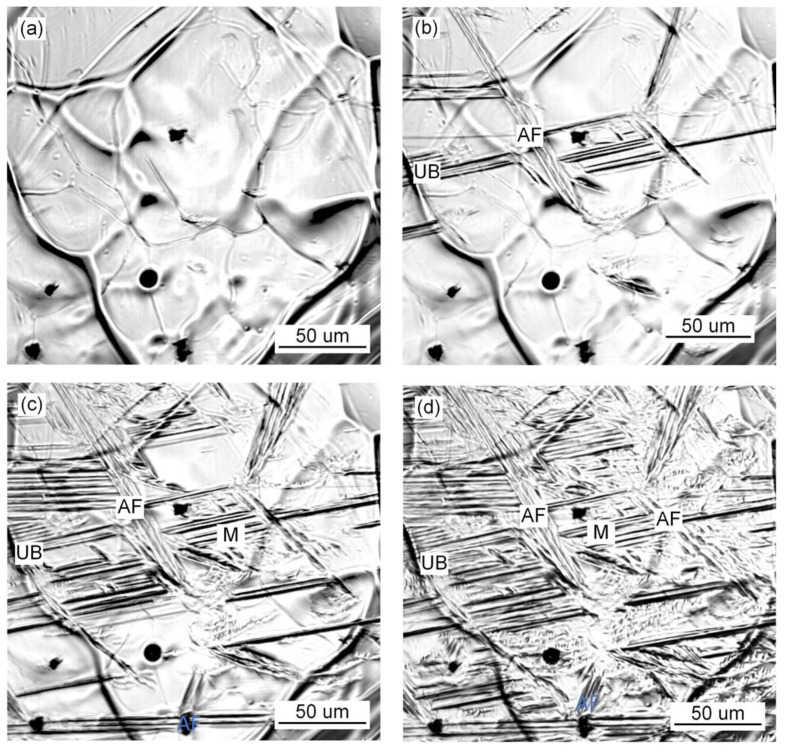
In situ observation of growth process of ferrite, bainite and martensite at 2% Pr_6_O_11_: (**a**) 408.47 s, 573 °C; (**b**) 411.86 s, 552.8 °C; (**c**) 414.25 s, 540.9 °C; (**d**) 418.84 s, 522.6 °C.

**Figure 10 materials-15-01257-f010:**
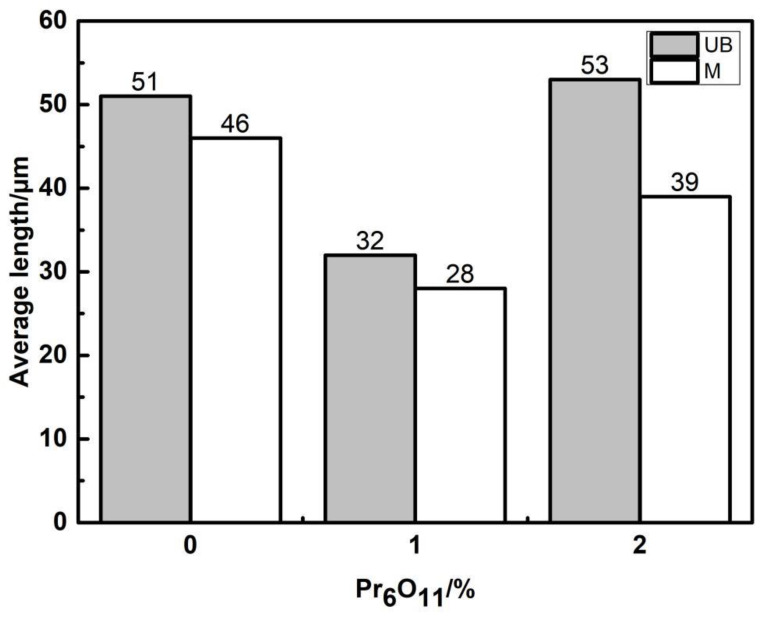
The average length of plates of UB and martensite.

**Figure 11 materials-15-01257-f011:**
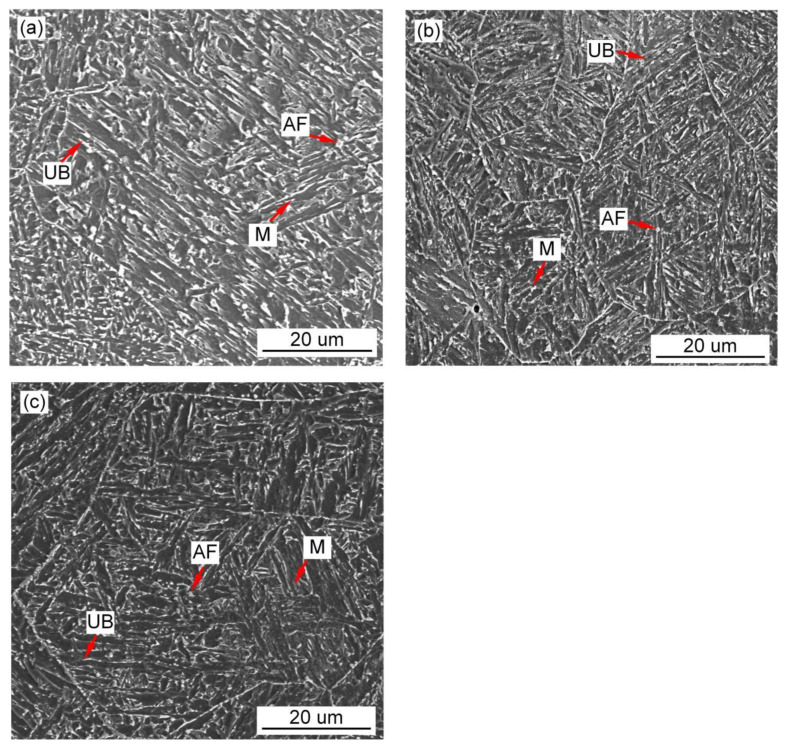
SEM images of microstructure of deposited metals with different Pr_6_O_11_ content: (**a**) 0% Pr_6_O_11_; (**b**) 1% Pr_6_O_11_; (**c**) 2% Pr_6_O_11_.

**Figure 12 materials-15-01257-f012:**
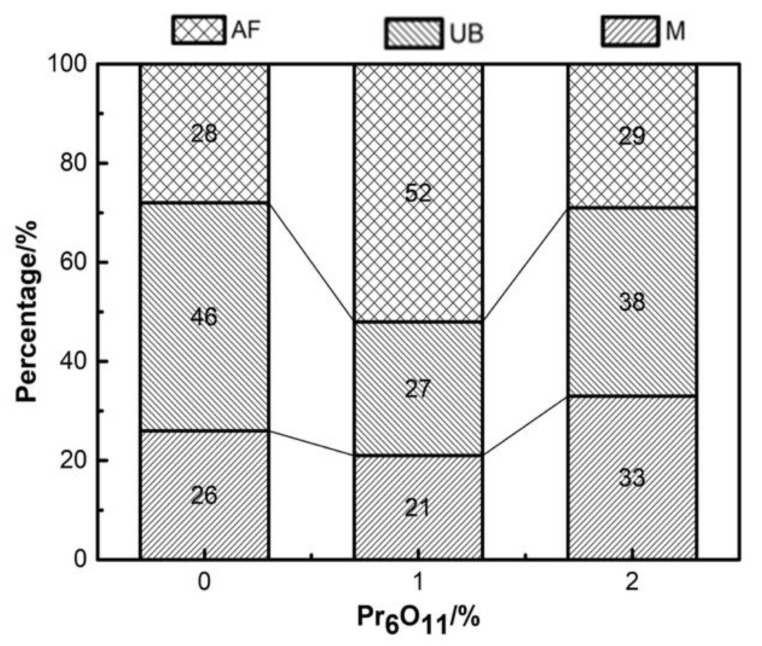
Quantitative statistical diagram of deposited metals with different Pr_6_O_11_ content.

**Figure 13 materials-15-01257-f013:**
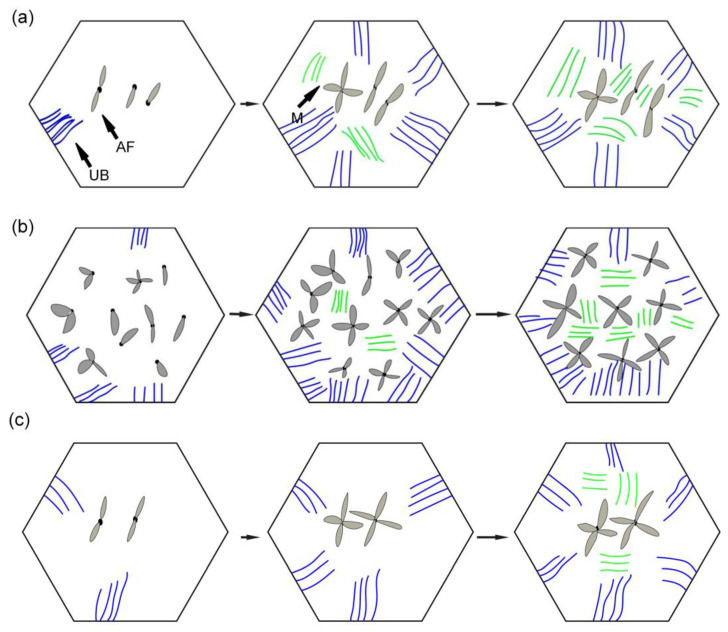
Schematic diagram of growth process of ferrite, bainite and martensite with different Pr_6_O_11_ content: (**a**) 0% Pr_6_O_11_; (**b**) 1% 0% Pr_6_O_11_; (**c**) 2% 0% Pr_6_O_11_.

**Figure 14 materials-15-01257-f014:**
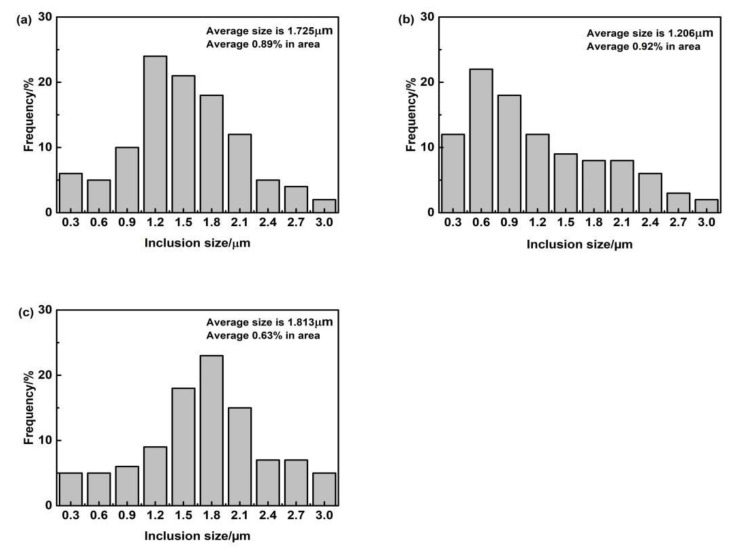
The size distribution of inclusions in deposited metals with different Pr_6_O_11_ content: (**a**) 0 % Pr_6_O_11_; (**b**) 1% Pr_6_O_11_; (**c**) 2% Pr_6_O_11_.

**Figure 15 materials-15-01257-f015:**
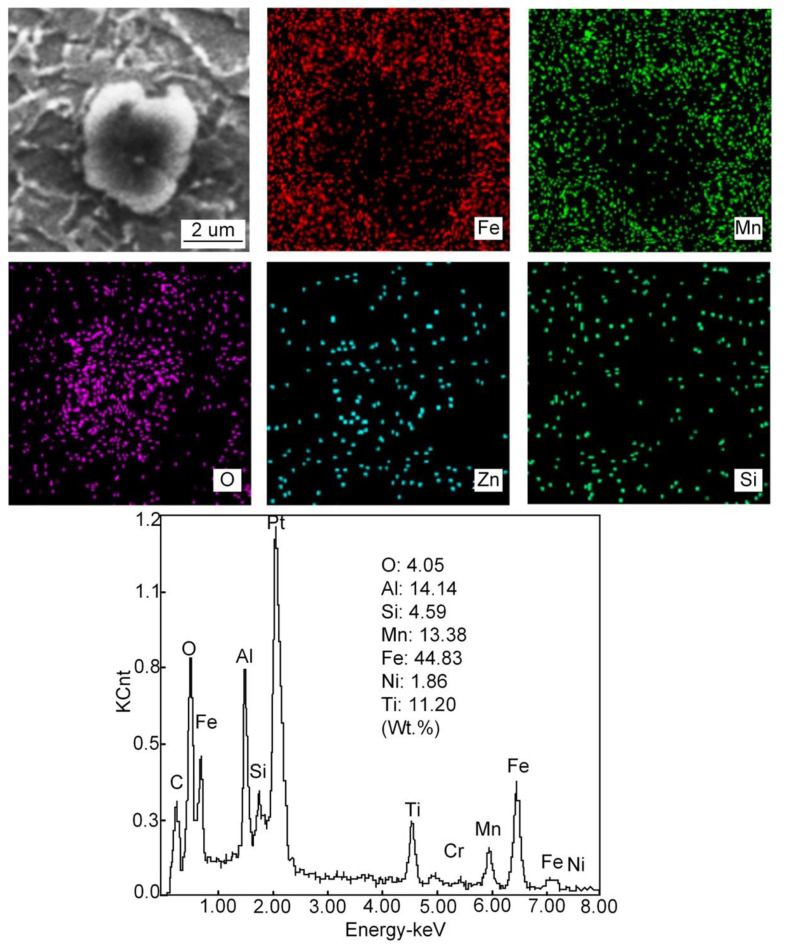
EDS analysis of inclusions in deposited metal at 0% Pr_6_O_11_.

**Figure 16 materials-15-01257-f016:**
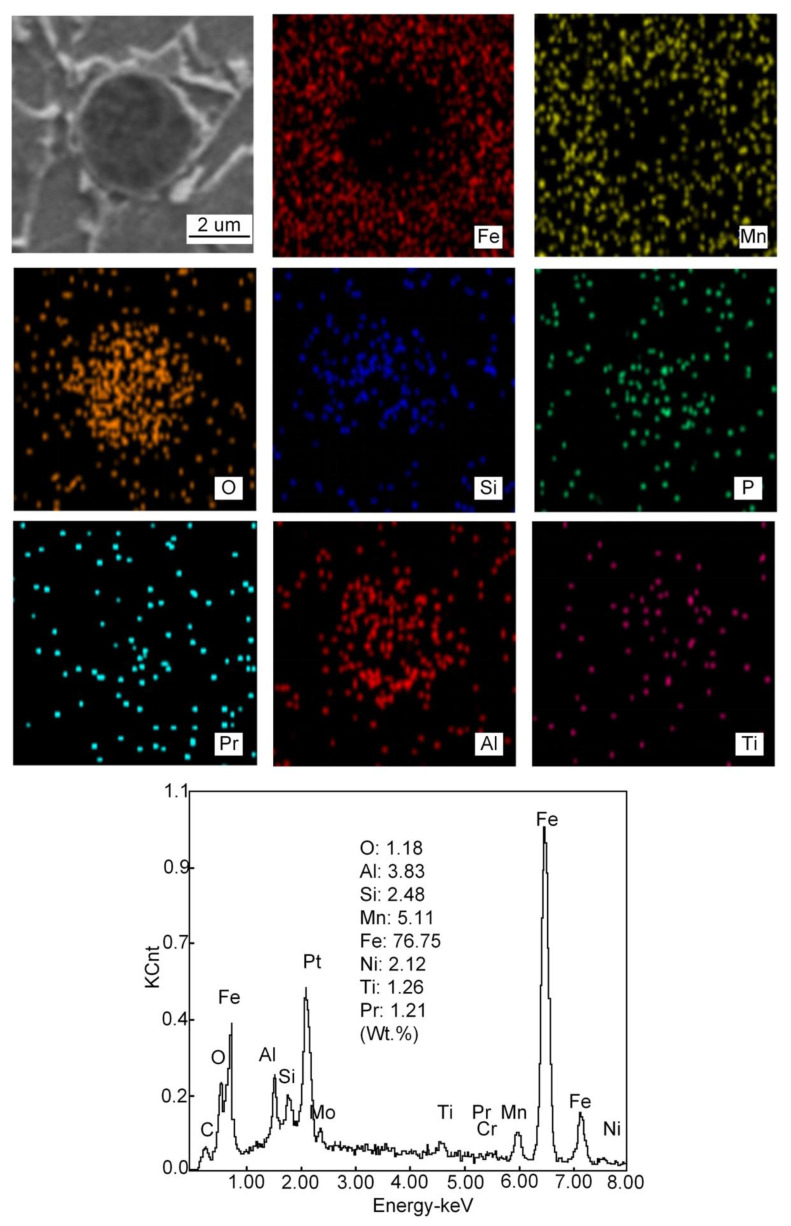
EDS analysis of inclusions in deposited metal at 1% Pr_6_O_11_.

**Figure 17 materials-15-01257-f017:**
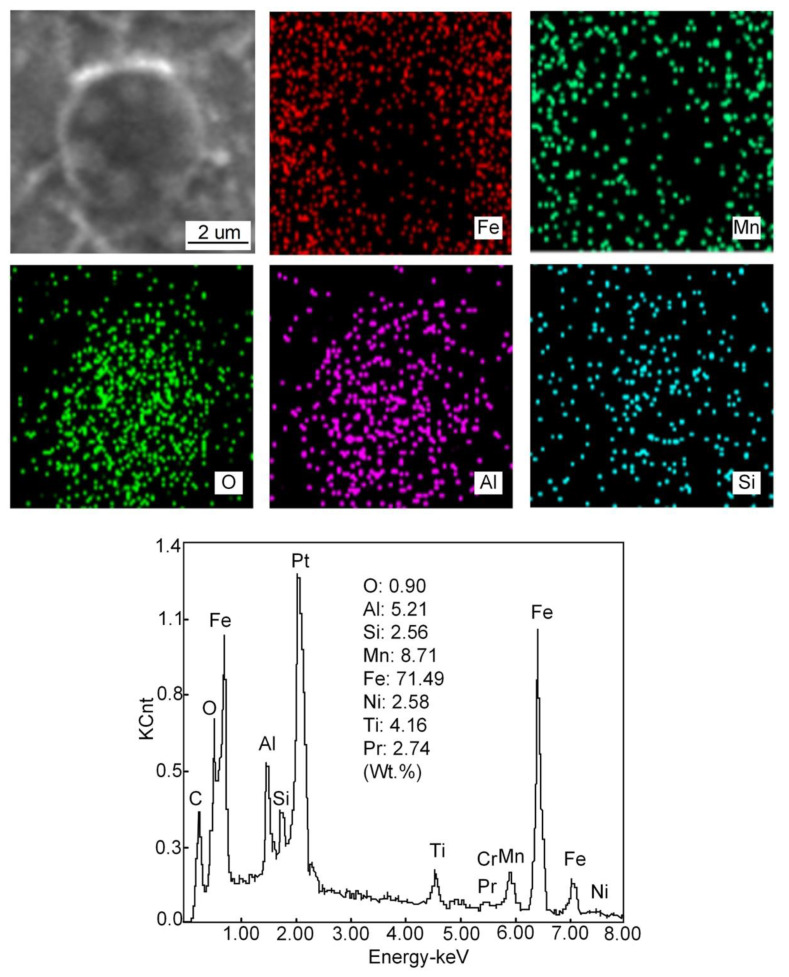
EDS analysis of inclusions in deposited metal at 2% Pr_6_O_11_.

**Figure 18 materials-15-01257-f018:**
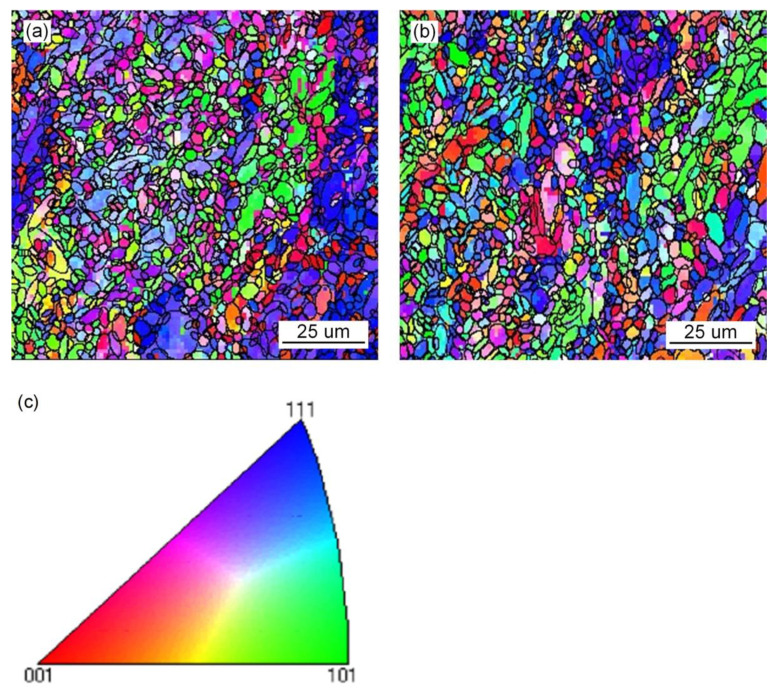
EBSD orientation maps of microstructure of deposited metals with different Pr_6_O_11_ content: (**a**) 0% Pr_6_O_11_; (**b**) 1% Pr_6_O_11_; (**c**) orientation color key.

**Figure 19 materials-15-01257-f019:**
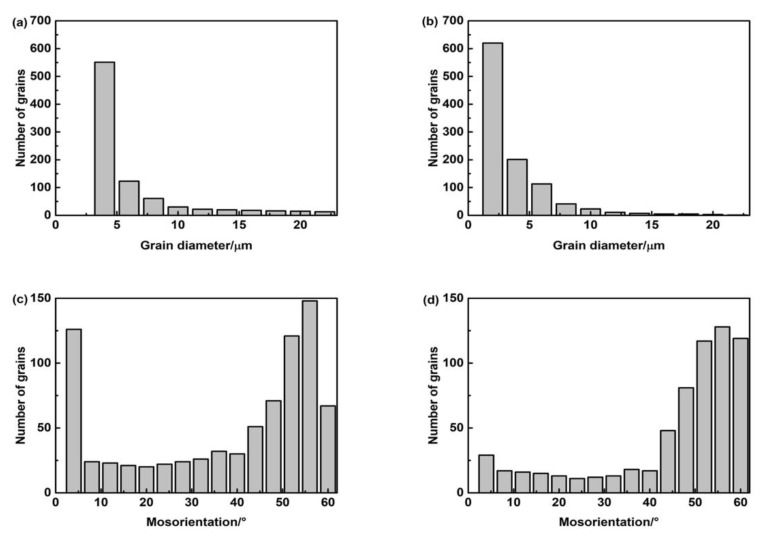
EBSD statistics of microstructure of deposited metals with different Pr_6_O_11_ content: (**a**) 0% Pr_6_O_11_ size distribution; (**b**) 1% Pr_6_O_11_ size distribution; (**c**) 0% Pr_6_O_11_ grain angle distribution; (**d**) 1% Pr_6_O_11_ grain angle distribution.

**Table 1 materials-15-01257-t001:** Welding parameters used in experiments.

Current /A	Voltage /V	Wire Stick-Out /mm	Shielding Gas /%	Gas Flow Rate /L·mm^−1^	Welding Speed /cm·min^−1^	Pre-Heat/Inter- Pass Temp. /°C
240	30	16	80 Ar + 20 CO_2_	20	28	150

**Table 2 materials-15-01257-t002:** Chemical composition of deposited metals (wt.%).

No.	C	Mn	Si	P	S	Cr	Ni	Mo	Pr	Al	Zn	Ti
1	0.038	1.891	0.424	0.013	0.008	0.405	2.122	0.625	0	0.011	0.0053	0.016
2	0.049	2.182	0.533	0.012	0.008	0.492	2.258	0.882	0.011	0.015	0.002	0.018
3	0.046	2.016	0.520	0.013	0.008	0.462	2.060	0.872	0.015	0.016	0.002	0.018

## Data Availability

The data presented in this study are available on request from the corresponding author.

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
