# Peer review of "In Situ Observation of Microstructural and Inclusions Evolution in High-Strength Steel Deposited Metals with Various Rare Earth Pr Contents"

_materials, 2022, doi:10.3390/ma15031257_

Round 1
Reviewer 1 Report
This study evaluates the effect of adding rare earth Pr content on the microstructure and inclusions of high strength steels. The study is very comprehensive and shows interesting findings on the effect of the rare earth metal. The manuscript is well-written and well-presented. I have some comments which should be addressed before publication of the manuscript, these can be found below.
- How was the chemical composition of the deposited metals measured?
- There seems to be a typo when describing the etchant, do the authors mean 5% Nital?
- Was the mechanical testing conducted to ASTM standards?
- Where does the Ti, Al & Zn come from in the inclusions as there is no measurement of these elements in the chemical composition (Table 2) of the deposited metals?
Author Response
Thank you for your time and effort in processing the manuscript.
We would like to express our sincere appreciations of your constructive comments concerning our article.
Based on them, we have made careful modifications on the original draft.
The point-by-point responses are in the attachment.

Reviewer 2 Report
Review
The submitted manuscript is interesting and contains new results. However, the discussion of the obtained results is very poor. Yet the manuscript should be dramatically improved before publication in the journal, according to the following comments.
- Introduction
- The aim of the article should be clarified in the text.
Section 2.
- Which (volume or weight) percent was used for the Pr6O11 content? Please clarify it.
- Which (atomic or weight) percent was used for characterization of the chemical composition? Please clarify it.
- Details of EBSD-analyses should be explained (step, voltage, etc.).
Section 3.1.
- The volume fraction and average size of inclusions should be estimated.
- Statistical analysis of grain size should be performed (Fig. 4).
- Engineering stress - strain curves and other tensile properties (elongation to failure, yield strength) should be presented and discussed.
Section 3.2.
- The reason why “AF in the grain boundaries first nucleates and grows on the inclusions” should be explained and discussed.
- The authors mention that “when 2% Pr6O11 is added, the inclusion nucleation particles are reduced due to the decrease of oxygen” (Page 7). It is a bit confusing. Please explain it in the text.
- The authors in the manuscript mention that the growth of UB and martensite is restricted due to dividing the austenite grains by AF. It will be better to estimate the average length of plates of bainitic ferrite and martensite and then compare.
- The effect of a decrease in the number of inclusions in 2%Pr6O11-alloy should be discussed in detail.
- It should be discussed why the obtained structures affect strength and toughness in a current way.
- It is so difficult to separate AF, UB, and martensite in Figure 8. The images should be made more contrast. The structures should be shown by arrows.
- Figure 10c is made in the distinctive colour mode in comparison to figures 10a and 10b.
- According to the text, the authors declare that “the cracks need to change many times when passing through this microstructure”. However, there are not any analyses of fracture after impact testing. It should be mentioned that fracture mechanisms were obtained. Furthermore, it seems possible that refinement of austenitic grain size resulted in the growth of toughness. Yet the effect of the obtained phase composition is not obvious.
- The statement “rare earth delays the transformation of bainite during in the welding and inhibits the formation of UB” was also suggested (Page 12), but any references or experimental results were not mentioned. Please explain it.
Section 3.3.
- The authors said that “the surface of inclusions is mainly Mn-Si oxide, and the inside is mainly Zn oxide” (Page 13). How is it shown that the outer layers are mainly Zn oxide? From the presented results, it is not clear.
- The average size of inclusions should be mentioned in figure 11 and discussed in the text.
- Why do the inclusions in the deposited metal change their shape? Please discuss it in detail.
Section 3.4.
- The authors write “reverse pole image” (Page 16), but “inverse pole image” is correct.
- How do the results of the structure characterization correlate with results obtained using other methods.
Author Response

(The authors gave the same response as above.)

Reviewer 3 Report
Although the favorable effect of the rare earth additions on the impact toughness of weld metal is confirmed, a question remains of why this effect weakens when the fraction of Pr oxide grows from 1 to 2%. Indeed, according to Figs. (9) and (10), the amount of acicular ferrite (AF) at 2% of this addition is almost the same as at its absence. Nonetheless, the impact toughness in the former case (2%) is still notably higher, although somewhat less than at optimal 1%. This fact hardly complies with the presumed DECISIVE role of AF. Evidently, the authors should also allow for other issues. For instance, if no alternative is proposed, one may consider the corresponding fractions of upper bainite and martensite as well as the dimensions of prior (austenite) grains whose boundaries could contain harmful segregations. The publication is recommended after the authors directly reply to this remark in the paper text.
Author Response
Thank you for your time and effort in processing the manuscript.
We would like to express our sincere appreciations of your constructive comments concerning our article.
Based on them, we have made careful modifications on the original draft.
The response is in the attachment

Round 2
Reviewer 2 Report
The manuscript can be accepted.